# Optically Transparent Metasurface Absorber Based on Reconfigurable and Flexible Indium Tin Oxide Film

**DOI:** 10.3390/mi11121032

**Published:** 2020-11-24

**Authors:** Lei Chen, Ying Ruan, Si Si Luo, Fu Ju Ye, Hao Yang Cui

**Affiliations:** College of Electronics and Information Engineering, Shanghai University of Electric Power, Shanghai 200090, China; chenlei@shiep.edu.cn (L.C.); ruanying@shiep.edu.cn (Y.R.); 19105093@mail.shiep.edu.cn (S.S.L.); 20171695@mail.shiep.edu.cn (F.J.Y.)

**Keywords:** optically transparent metasurface, reconfigurable metasurface, flexible metasurface, indium tin oxide film

## Abstract

In this paper, we present a flexible, breathable and optically transparent metasurface with ultra-wideband absorption. The designed double layer of indium tin oxide (ITO) films with specific carved structure realizes absorption and electromagnetic (EM) isolation in dual-polarization, as well as good air permeability. Under the illumination of x- and y-polarization incidence, the metasurface has low reflectivity and transmission from about 2 to 18 GHz. By employing ITO film based on polyethylene terephthalate (PET), the presented metasurface also processes the excellent flexibility and optically transparency, which can be utilized for wearable device application. In addition, the dual-layer design enables mechanically-reconfigurable property of the metasurface. The transmission and reflection coefficients in two polarizations show distinct difference when arranging the different relevant positions of two layers of the metasurface. A sample with 14*14 elements is designed, fabricated and measured, showing good agreement with the simulation results. We envision this work has various potentials in the wearable costume which demands both EM absorption and isolation.

## 1. Introduction

Over recent decades, metamaterials have attracted great attention due to their remarkable and artificially-designable electromagnetic (EM) properties [1], which do not exist in nature. For example, the permittivity and permeability of the artificial structure can be designed into zero [2] or even negative values [3]. Such unreal characteristics have aroused various devices like perfect lens [4], cloaking [5,6] and illusion carpets [7]. To address the processing limits of three-dimensional (3D) structures, metasurfaces and metafilms with ultrathin profile and similar manipulation capability on EM waves are presented [8,9,10]. Not only the reflection phase [11,12,13], but also the amplitude [14,15] and polarization [16] can be arbitrarily modulated. Benefiting from these advantages, metasurfaces have facilitated diverse EM phenomena such as anomalous refraction [17], vortex beams [18,19] and hologram [20].

Among these splendid applications, absorption based on metamaterials has always been a hot research field, which occupies important status in various engineering for EM shielding and isolation. Many researches on metasurface absorbers realize absorption by designing special metal structure on insulated substrate such as epoxy glass cloth [21,22,23] (such as FR4, a kind of epoxy glass fiber cloth substrate), reinforced hydrocarbon/ceramic laminates [24,25] and so on. The operating frequency range of the metamaterial absorber has covered microwaves [26], terahertz [27] and optical domains [28,29]. Other methods on frequency domain focus on realizing multi-frequency [30,31] or wideband absorption [32,33]. Besides, other substrate materials like water [34,35], liquid crystal [36] and other semiconductor substrates [37] are also applied to achieve tunable absorption performance.

However, the vast majority of absorber designs are based on rigid and airtight substrate. Such metasurfaces can hardly be used for wearable applications, which requires the metasurface for both air permeability and flexibility, as well as ultra-low profile and weight. To explore the solution of these drawbacks, we present an optically-transparent, flexible absorbing metasurface with a breathable structure. By utilizing dual layers of the metasurface, the propagating wave in two orthogonal polarizations can be absorbed. The absorption performance can be changed by applying a different mechanical position. In addition, with the carved structure on the indium tin oxide (ITO) thin films, we achieve good air permeability and flexibility, as well as good EM shielding to transmission wave, which has great potential for EM-dissipation wearable clothes and devices. The measured results show great results with the simulation design, which validates the feasibility.

## 2. Design and Results

Figure 1 exhibits the schematic of the optically transparent metasurface based on the ITO thin films. By sculpturing the special carved structure on the thin but lossy films, we simultaneously achieve good absorption for both linear and perpendicular polarization (along x- and y-axis) and air-permeability on the metasurface. The loss resistance rate endowed by the ITO film brings considerable energy consumption for EM propagating wave. Benefiting from the ultra-thin thickness of the polyethylene terephthalate (PET), the metasurface possesses excellent flexibility and is lightweight. More interestingly, the ITO film and PET substrate also provide high optical transparency for visible light. As depicted in Figure 1, we pile up two metasurface layers with an orthogonal angle to obtain the absorption in both x- and y-polarizations.

In Figure 2, we present the designed structure of the metasurface element, which is composed of two layers of PET substrate, as shown in Figure 2a. Ultrathin ITO films (with about 13 nm thickness) are covered on the PET substrate, with 0.3 mm thickness. Figure 2b provides the front view of the element, with the designed size parameters labeled on it. The detailed dimensions of the unit cell in the figure are given as following: *a* = 15 mm, *b* = 14 mm, *c* = 5 mm, *d* = 1.5 mm, *h* = 1 mm, *t* = 0.3 mm, where *a* is periodic of the element, *h* is the air gap between two PET layers, *t* is the thickness of the single layer of PET substrate (a dielectric constant of 4.5 and loss tangent of 0.005). The numerical simulations are performed on a commercial software, CST Microwave Studio. The periodic boundary condition is applied in simulation to simulate an ideal metasurface with infinite size. To effectively analyze the EM absorption characteristics, we calculate the S-parameters of the metasurface element in simulations. The S-parameter here refers to the 2-port scattering parameters of the metasurface element. Please note that port 1 and port 2 are, respectively, set at the two sides of the unit, along the z-axis in Figure 2a, to test the reflection on both sides. S11 and S22 mean the reflection coefficients of the port 1 and 2. S21 refers to the transmission coefficients from port 1 to port 2; S12 refers to the transmission coefficients from port 2 to port 1. Figure 2c presents the simulated reflection coefficients (S11 and S22) marked in blue and pink colors, respectively. Since the element structure is symmetrical in x- and y-direction, here we give the results of reflection coefficients S11 and S22, which are exactly equivalent to the two orthogonal polarization for port 1. We notice that the simulated S11 and S22 are all below about -15 dB from 2 to 18 GHz, implying good absorption for the reflection wave. For the transmission wave, the simulation data is listed in Figure 2d. The transmission coefficients S21 and S12 are almost the same, because of this passive reciprocal system. The EM shielding band below −10 dB starts from 5.4 GHz to 18 GHz. Comparing to the reflection absorbing, the transmission isolation in the low-frequency band is a little weak. This problem can be further improved by overlaying more layers of the metasurface or applying ITO films with higher resistance. To clearly present the absorption performance, we calculated the normalized absorbing coefficient in Figure 2e, when the excitation is set at port 1 (from the front side) and port 2 (from the backside), respectively. We can clearly observe that the absorption from 2 to 18 GHz is almost 0.7, in which the performance in high frequency is much better than it in low frequency. We also show the simulated electric field distribution on the element structure when the excitation is port 1 or port 2. Both the front and the back view is exhibited in Figure 2f,g, to clearly show its resonance. When the excitation is set at port 1, the electric field on the front layer is obviously stronger than the back layer; on the contrary, when the port 2 is excited, the electric field on the backside is much stronger.

In order to exhibit the influence of different parameters on the electromagnetic performance of the element, the parameters of h (the air gap between two PET layers) and R (the resistance of ITO films) were scanned. Figure 3 shows the variation of reflection coefficient S11 and transmission coefficient S21 under different polarization modes when h changes. Figure 3a,b express the variation of S11 with h under x and y polarization, respectively. As shown in Figure 3a, the maximum reflection energy occurs when h = 1 mm, and the minimum reflection energy occurs when h = 5 mm. With the increase of h, S11 decreases gradually. Figure 3b indicates the variation of reflection coefficient S11 with h under y polarization. Obviously, the change of h has no significant effect on S11, and the curve of S11 almost overlaps with the change of h. In particular, no matter what the value of h and whether the boundary condition of the unit cell is set as x-polarization or y-polarization, the reflection coefficient S11 is less than −15 dB in the frequency range of 2 GHz to 18 GHz, which achieves good absorbing effect on reflected wave. The change of transmission coefficient S21 with h under x polarization and y polarization is shown in Figure 3c,d, respectively. By comparing Figure 3c,d, it can be found that the variation curve of the transmission coefficient S21 is almost the same under different boundary conditions. When h increases, the transmission coefficient S21 increases gradually. When h = 5 mm, the transmission energy is the largest, and the transmission energy is the smallest when h = 1 mm.

Figure 4 shows the changes of reflection coefficient S11 and transmission coefficient S21 with different ITO film resistance values R in different polarization modes. Figure 4a,b indicate the variation of reflection coefficient S11 with R under x and y polarization, respectively. According to the analysis of Figure 4a, in the frequency range of 2 to 18 GHz, when R = 3, 10, 30, 50 or 100 Ω/sq, S11 is greater than −10 dB, which indicates the fair performance of the absorption effect. When R = 300 Ω/sq, S11 is less than −18 dB, which can achieve a better absorbing effect in a wide frequency range. Similarly, it can be found from Figure 4b that no matter in x-polarization or y-polarization, only when R = 300 Ω/sq, S11 is less than −15dB in the frequency range of 2 GHz to 1 GHz and the element can achieve good absorption of reflected wave. For the transmission coefficient S21, the variation at x polarization and y polarization is shown in Figure 4c,d. Because the unit cell is a passive reciprocity system, S21 is almost the same in different polarization modes. With the increase of R, the bandwidth of S21 which is less than −10 dB increases gradually. When R = 300 Ω/sq, S21 is less than −10 dB in the frequency range of 5.4 to 18 GHz, realizing broadband electromagnetic shielding.

In addition to the above characteristics, the metasurface we designed also has mechanical reconfigurability. We can adjust the relative position of the element, such as translating the element along the x-axis or y-axis, to change the reflection and transmission performance of the unit cell. In Figure 5a, the relative position of two layers metasurface is translated along x-axis with a half period, and in Figure 5b, the relative position of two layers is translated along x-axis with a half period. Simulation coefficients of elements under different translation modes are described in Figure 5c,d. Among them, dark blue and light blue indicate the reflection coefficient S11 of the unit cell at x polarization and y polarization, respectively. Rose-red and bright red express the transmission coefficient S21 of the element at x polarization and y polarization, respectively. Figure 5c shows the simulation coefficient of the element after a half period (the period is equal to 15 mm) along the x-axis. Compared with the results in Figure 2c,d which are before translation, the energy of S11 in x-polarization is much lower than −20 dB in the frequency range of 2 to 18 GHz, while the S11 in y-polarization has a similar response as before. And S21 is below about −10 dB from 7 to 16 GHz, whose bandwidth is less than before. The simulation coefficient after the element is shifted along x-axis and y-axis for a half period is shown in Figure 5d. Compared with Figure 2c,d which are before translation, S11 has no significant change. Furthermore, in the frequency range from 10 to 18 GHz, S21 is increased overall of about 5 dB.

The measurement is performed within a support platform with two broadband horn antennas, as illustrated in Figure 6a. The metasurface sample is fixed on the central support to test its transmission and reflection coefficients. Two broadband antennas with an operating band from 2 to 18 GHz is set at the distance of 300 mm away from the metasurface sample. The reflection and transmission coefficients are measured by a vector network analyzer (Agilent 8722ET), which is connected to transmitting and receiving horns. It should be noted that the support plane contains PEC board with the microwave absorbing materials covering on it, to prevent unnecessary reflection and transmission. The fabricated metasurface sample is exhibited in Figure 6b,c, which, respectively, relates to the single layer metasurface sample and assembled two-layer metasurface sample. The ITO metasurface is fabricated using magnetron sputtering technology, which has become a mature process for ITO film fabrication due to large fabrication size and great homogeneity of film growth. The metasurface sample is processed using a laser sculpture machine (WF-UV-W3/5) to produce the designed structure on a whole PET film. About 300 Ω/sq ITO films are built on the 0.3 mm PET films, whose dielectric constant and loss tangent are 4.5 and 0.005, respectively. For such a thickness of the ITO-PET film, about 88% optical transparency is achieved.

In experiment verification, the S-parameter of the reflection and transmission are measured, as shown in Figure 7. The theoretical data is also provided for a clear comparison. The S-parameter here refer to the 2-port scattering parameters measured by a vector network analyzer. The reflection coefficients of the front and back side of the metasurface are listed in Figure 7a,b, respectively. S11 and S22 means the reflection coefficients of the port 1 and 2. We can observe that the reflection amplitude is below −16 dB from 2 to 18 GHz, which show good coincidence with the simulated data. The presented results fully demonstrate the good absorption performance and ultra-wideband for the reflected wave. To verify the EM isolation property, the transmission coefficients are also measured. S21 refers to the transmission coefficients from port 1 to port 2; S12 refers to the transmission coefficients from port 2 to port 1. The results are provided in the Figure 7c,d, marked in blue color. According to the measured curves, the transmission power is below −10 dB from about 2 GHz to 18 GHz, also suggesting good performance and agreement with simulations. It should be noted that the data in Figure 7c,d is almost the same since the presented metasurface as well as the measurement is a reciprocal system. Thus the transmission coefficients S21 and S12 are the theoretically same. The slight errors observed from the results are mainly resulted from the following reasons: (1) the fabrication error of the metasurface films; (2) the slight crispation of the metasurface sample in the measurement. (3) the errors from the manual operations. More specifically, in Figure 7a,b, the measured S11 is a little higher than the theoretical value in high frequency range (about 10–18 GHz), while the measured S22 is a little lower than the theoretical value in high frequency range. This phenomenon may be due to the error of fabrication precision. For transmission results, the measured results are overall a little lower than the theoretical data, which may be resulted from the imperfect plane wave generated from the horn antenna. The spherical wave produces extra energy loss in measurement.

## 3. Discussion

In summary, we present an optically-transparent metasurface absorber based on ITO films to realize a flexible and ultra-wideband absorption for dual-polarization. The specific carved structure on the PET substrate (with ITO films on it) promises good air permeability. An ultra-wide absorbing band is achieved from 2 to 18 GHz, simultaneously with good flexibility and optical transparency. By overlaying two metasurfaces, both x- and y-polarization wave can be absorbed. Comparing to the previous optically transparent metasurface for absorption, our design first combine flexibility, reconfigurability, breathability and wideband absorption into one metasurface, suggesting great composite functionality. A dual-layer metasurface sample with 14*14 elements is designed, fabricated and measured to demonstrate its performance. The measured results show great agreement with our design, fully verifying the results. The presented metasurface combines optically transparency, reconfigurable microwave absorption and excellent flexibility in a single and ultrathin design. Such functionalities can not only find their potential applications in the conventional domain like EM communication [38], imaging and display [39]; but also biological sensing and interaction [40], intelligent internet of things [41]. Therefore, we believe this work may pave a new method of intelligent wearable devices for EM absorbing and shielding.

## Figures and Tables

**Figure 1 micromachines-11-01032-f001:**
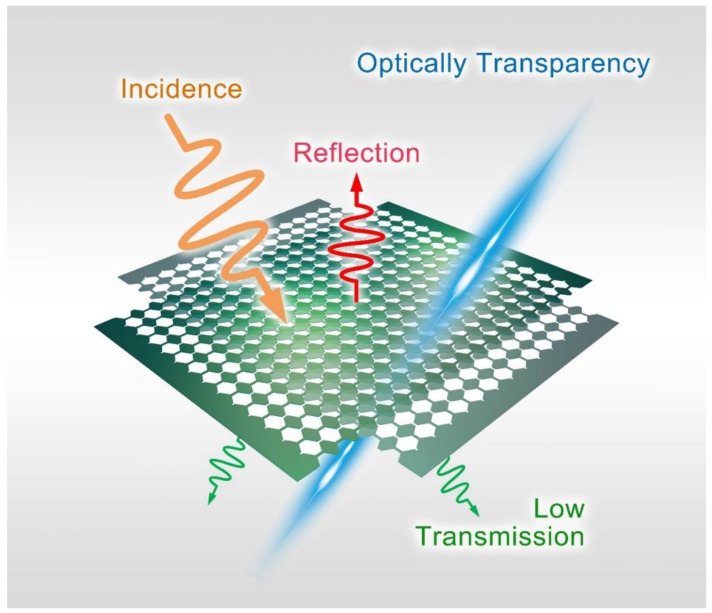
The schematic diagram of the presented metasurface absorber. For dual-polarization incidence, the metasurface achieves both low reflection and transmission. By utilizing the ITO carved films, good flexibility, air permeability and flexibility are realized simultaneously.

**Figure 2 micromachines-11-01032-f002:**
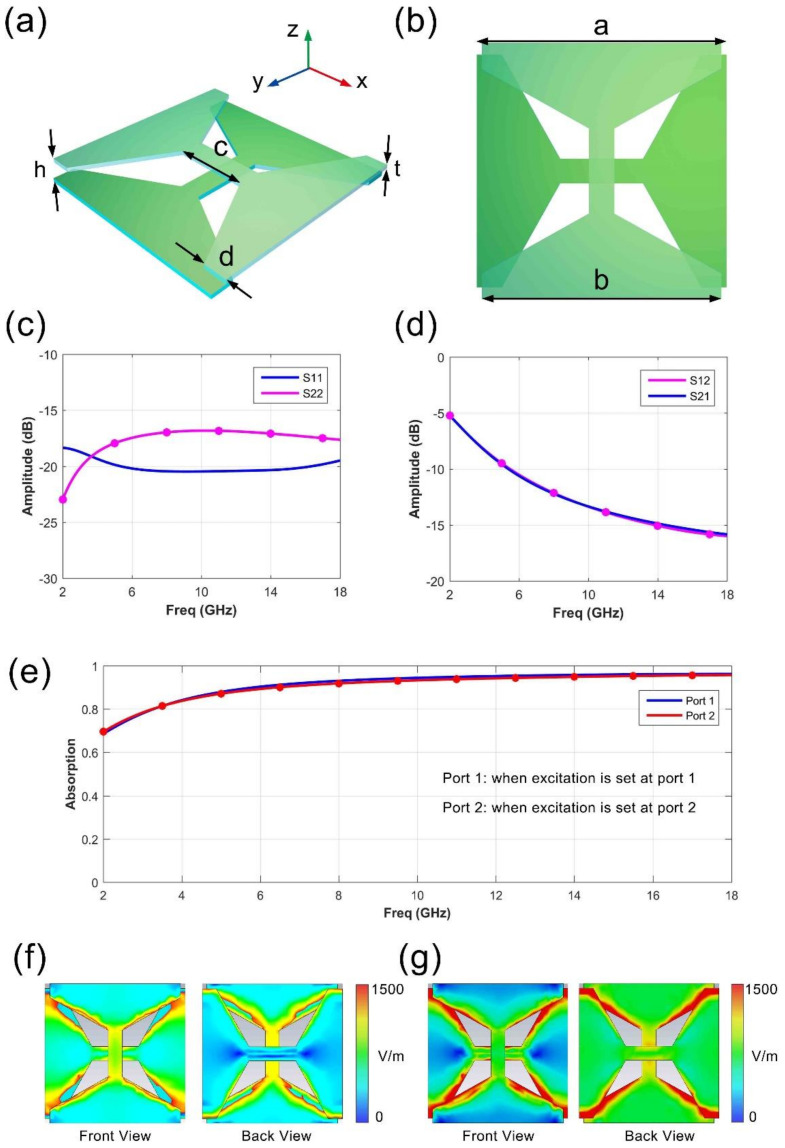
The designed element structure and the simulated results. (**a**) The detailed structure of the dual-layer metasurface. (**b**) The front view of the presented element. (**c**,**d**) The simulated reflection and transmission data of the element. (**e**) The absorption results when excitation is set at port 1 and 2. (**f**,**g**) The electric field distribution of the element when excitation is set at port 1 and 2, in which both the front and the back view is provided.

**Figure 3 micromachines-11-01032-f003:**
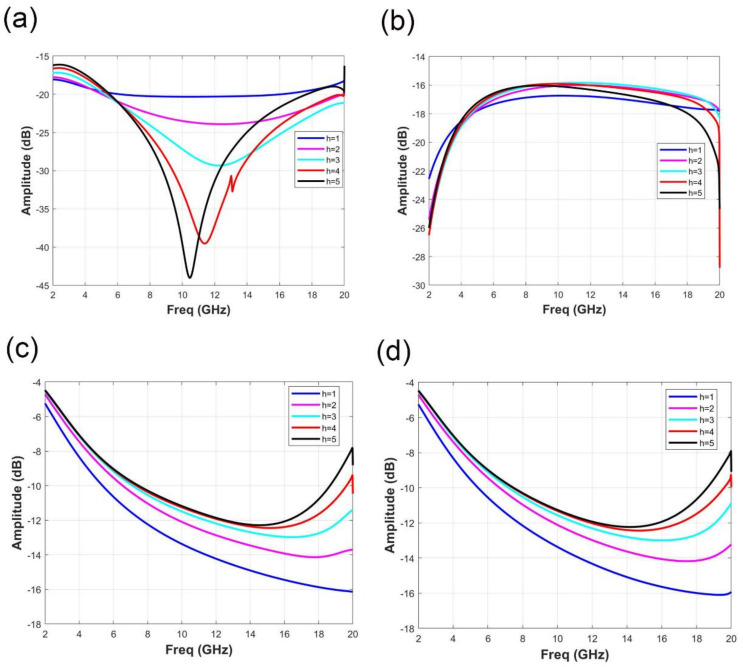
Variation of reflection coefficient S11 and transmission coefficient S21 under different polarization modes when h changes. (**a**) Under x polarization, the reflection coefficient S11 changes with h. (**b**) Under y polarization, the reflection coefficient S11 changes with h. (**c**) Under x polarization, the transmission coefficient S21 changes with h. (**d**) Under y polarization, the transmission coefficient S21 changes with h.

**Figure 4 micromachines-11-01032-f004:**
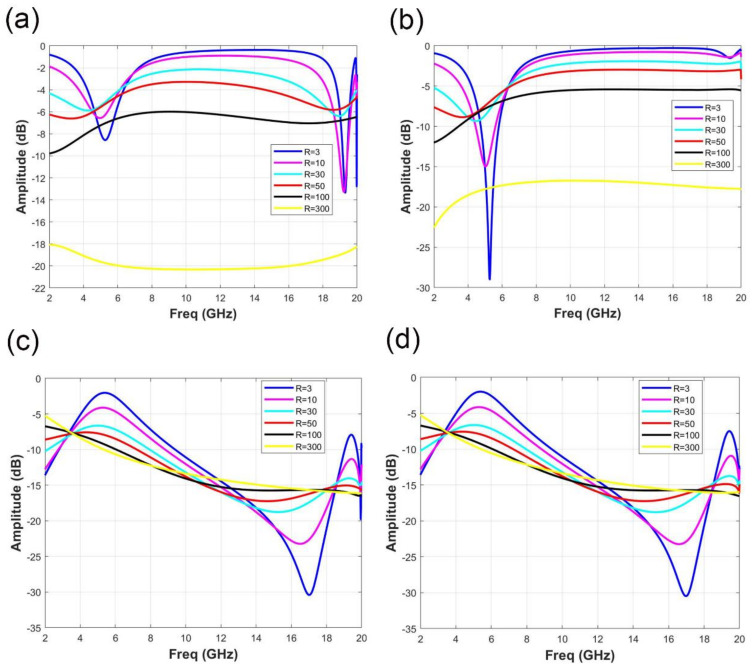
Variation of reflection coefficient S11 and transmission coefficient S21 under different polarization modes when R changes. (**a**) Variation of the reflection coefficient S11 under x polarization when R changes. (**b**) Variation of the reflection coefficient S11 under y polarization when R changes. (**c**) Variation of transmission coefficient S21 under x polarization when R changes. (**d**) Variation of transmission coefficient S21 under y polarization when R changes.

**Figure 5 micromachines-11-01032-f005:**
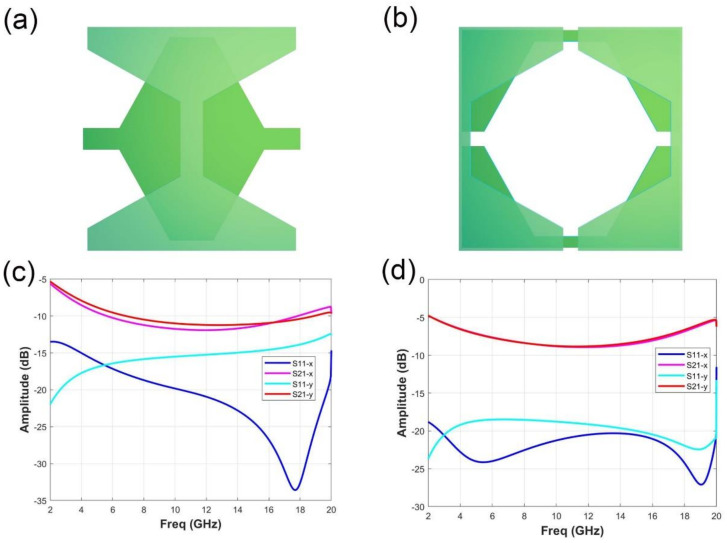
Simulation coefficients of elements under different translation modes. (**a**) The view of the element when translated along the x axis for 1/2 period. (**b**) The view of the element after 1/2 period translation along x-axis and y-axis. (**c**) The simulation coefficient after the element is translated along the x axis for 1/2 period. (**d**) Simulation coefficient of the element after 1/2 period translation along x-axis and y-axis.

**Figure 6 micromachines-11-01032-f006:**
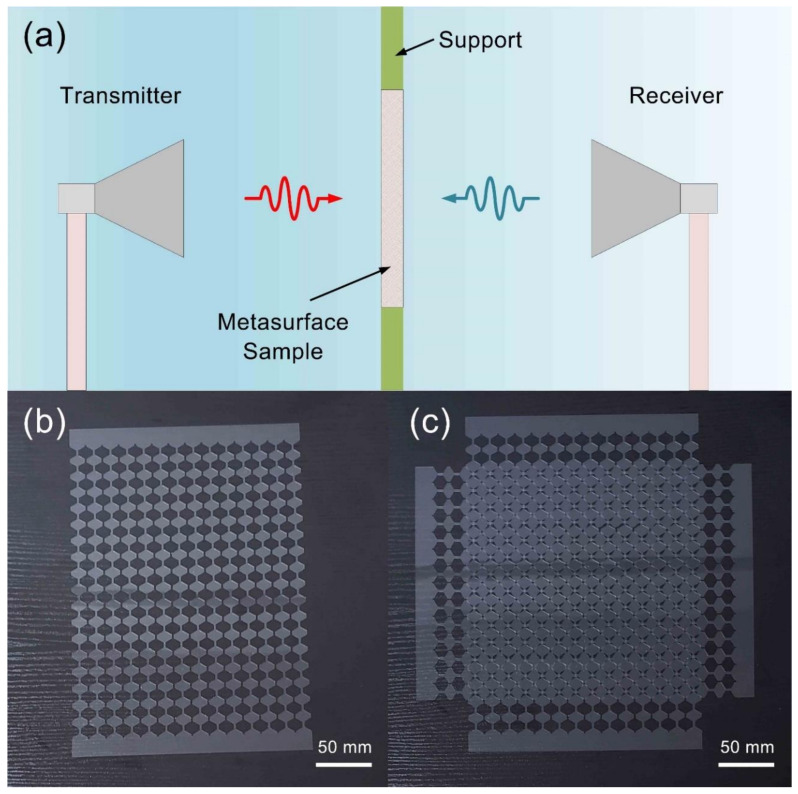
The experiment configuration of the presented metasurface and the fabricated metasurface sample. (**a**) The measurement performed within a wide-band horn antenna platform to test the S-parameters. (**b**) The fabricated metasurface sample of the single layer. (**c**) The assembled dual-layer metasurface absorber.

**Figure 7 micromachines-11-01032-f007:**
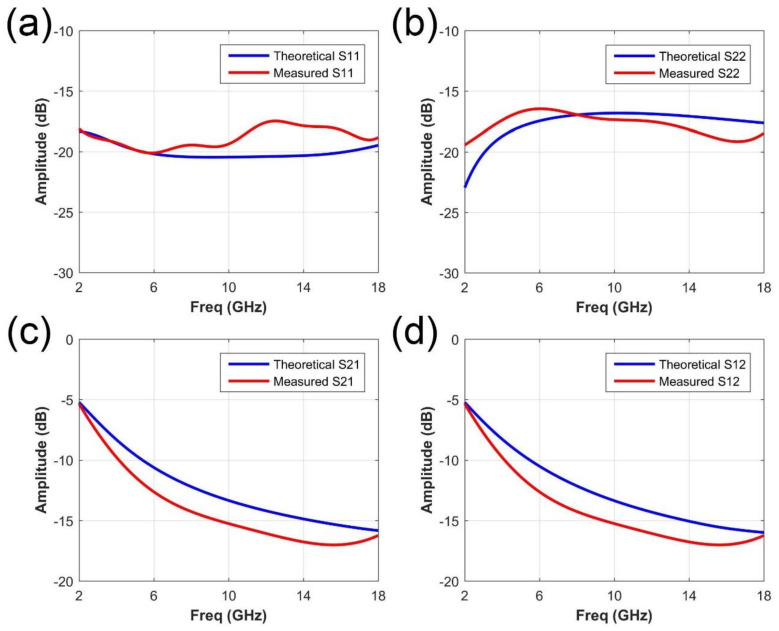
The measured reflection and transmission coefficients of the presented metasurface. (**a**,**b**) The reflection coefficients of two ports, S11 and S22, measured in the experiments. (**c**,**d**) The transmission coefficients, S21 and S12, measured in the experiments.

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
