# Peer review of "Optically Transparent Metasurface Absorber Based on Reconfigurable and Flexible Indium Tin Oxide Film"

_micromachines, 2020, doi:10.3390/mi11121032_

Round 1

Reviewer 1 Report

Overall an excellent research article. There are minor grammar issues that need to be fixed. Please proof read the manuscript carefully.

To improve the quality of the article I would like to suggest to add specific details of the experimental set up. For example, what instruments or methods were used to fabricate meta-surface samples, and how the experimental data were collected for reflection and transmission.

The distinct between the simulation results and measured experimental results is not clear for some of the graphs. Make sure to differentiate the two and intentionally point it out.

Lines 195-197 provide a general explanation for the disagreement for theoretical and measured data. Although the authors label them as “slight errors” the difference between the experimental and theoretical values are significant and require a detailed explanation. At least for the trends observed for figures 7-c and d.

The discussion section of the paper does not complement interesting findings of the research paper and article. That is to say the discussion seems to be weak compared to the rest of the article.  A meaningful discussion (rather than just summing up the paper) may further improve the quality of the article.

Author Response

To Reviewer 1

Comment:

To improve the quality of the article I would like to suggest to add specific details of the experimental set up. For example, what instruments or methods were used to fabricate meta-surface samples, and how the experimental data were collected for reflection and transmission.

Our Response:

Thank you very much for your good suggestions. We have added more details about measurement and fabrication. The ITO metasurface is fabricated using magnetron sputtering technology, which has become a mature process for ITO film fabrication [R1] due to large fabrication size and great homogeneity of film growth. This process contains a series of machines like film plating machine, vacuum air pumping, sample injector, and the related control cabinet. The structure sculpture of the metasurface is performed by a laser sculpture machine (WF-UV-W3/5) with high precision. In the measurement, the reflection and transmission coefficients are measured by a vector network analyzer (Agilent 8722ET), which connected to transmitting and receiving horns.

We have added the above information into the revised manuscript as following,

The supplemented sentences:

The reflection and transmission coefficients are measured by a vector network analyzer (Agilent 8722ET), which connected to transmitting and receiving horns.

The ITO metasurface is fabricated using magnetron sputtering technology, which has become a mature process for ITO film fabrication due to large fabrication size and great homogeneity of film growth. The metasurface sample is processed using laser sculpture machine (WF-UV-W3/5) to produce the designed structure on a whole PET films.

[R1] Y. Shen, Z. C. Feng, and H. Y. Zhang, "Study of indium tin oxide films deposited on colorless polyimide film by magnetron sputtering," Materials & Design, Article vol. 193, p. 7, Aug 2020.

Comment:

The distinct between the simulation results and measured experimental results is not clear for some of the graphs. Make sure to differentiate the two and intentionally point it out.

Our Response:

Thank you very much for your good comments. To clearly differentiate some similar curves in some graphs, we have revised the curve plotting style in Fig. 2. Besides, we find the simulation and measurement data in Fig. 6 are all clear, so we remain its current curve style in the figure. It should be noted that the data in Fig. 7(c) and (d) is almost the same since the presented metasurface as well as the measurement is a reciprocal system. Thus the transmission coefficients S21 and S12 are the theoretically same. The difference between simulation and measurement data is mainly resulted from the error in fabrication and measurement, which has been discussed in the paper. We have supplemented the above point in the revised paper as below,

The supplemented sentences:

It should be noted that the data in Fig. 7(c) and (d) is almost the same since the presented metasurface as well as the measurement is a reciprocal system. Thus the transmission coefficients S21 and S12 are the theoretically same.

Comment:

Lines 195-197 provide a general explanation for the disagreement for theoretical and measured data. Although the authors label them as “slight errors” the difference between the experimental and theoretical values are significant and require a detailed explanation. At least for the trends observed for figures 7-c and d.

Our Response:

Thank you very much for your good suggestion. As we have discussed in original paper, the error between the theoretical and measured results is mainly due to the fabrication and measurement error. More specifically, in Fig. 7(a) and 7(b), the measured S11 is a little higher than theoretical value in high frequency range (about 10-18GHz), while the measured S22 is a little lower than theoretical value in high frequency range. This phenomenon may be due to the error of fabrication precision. For transmission results, the measured results are overall a little lower than the theoretical data, which may be resulted from the imperfect plane wave generated from the horn antenna. The spherical wave produces extra energy loss in measurement.

We have supplement the above discussion in the revised paper as following,

The supplemented sentences:

More specifically, in Fig. 7(a) and 7(b), the measured S11 is a little higher than theoretical value in high frequency range (about 10-18GHz), while the measured S22 is a little lower than theoretical value in high frequency range. This phenomenon may be due to the error of fabrication precision. For transmission results, the measured results are overall a little lower than the theoretical data, which may be resulted from the imperfect plane wave generated from the horn antenna. The spherical wave produces extra energy loss in measurement.

Comment:

The discussion section of the paper does not complement interesting findings of the research paper and article. That is to say the discussion seems to be weak compared to the rest of the article.  A meaningful discussion (rather than just summing up the paper) may further improve the quality of the article.

Our Response:

Thank you very much for your good suggestion. The presented metasurface combines optically transparency, reconfigurable microwave absorption and excellent flexibility in a single and ultrathin design. Such functionalities can not only find their potential applications in conventional domain like EM communication [R2], imaging and display [R3]; but also biological sensing and interaction [R4], intelligent internet of things [R5]. Therefore, we believe this work may pave a new method of intelligent wearable devices for EM absorbing and shielding.

To enrich the discussion, we have added the above discussion in the revised paper as below,

The supplemented sentences:

The presented metasurface combines optically transparency, reconfigurable microwave absorption and excellent flexibility in a single and ultrathin design. Such functionalities can not only find their potential applications in conventional domain like EM communication [R2], imaging and display [R3]; but also biological sensing and interaction [R4], intelligent internet of things [R5]. Therefore, we believe this work may pave a new method of intelligent wearable devices for EM absorbing and shielding.

[R2] T. J. Cui, S. Liu, G. D. Bai, and Q. Ma, "Direct transmission of digital message via programmable coding metasurface," Research, vol. 2019, p. 2584509.

[R3] X. Zou et al., "Imaging based on metalenses," PhotoniX, vol. 1, no. 1, p. 2, 2020/03/04 2020

[R4] O. A. Araromi et al., "Ultra-sensitive and resilient compliant strain gauges for soft machines," Nature, vol. 587, no. 7833, pp. 219-224, 2020

[R5] J. Gubbi, R. Buyya, S. Marusic, and M. Palaniswami, "Internet of Things (IoT): A vision, architectural elements, and future directions," Future Generation Computer Systems-the International Journal of Escience, vol. 29, no. 7, pp. 1645-1660, Sep 2013

Reviewer 2 Report

In the paper, the authors experimentally demonstrate an optically transparent absorber with broadband absorption in the microwave region. The experimental measurements agree well with the theoretical predictions, showing broadband absorption. Though the topic has been widely investigated, the paper is still interesting. Therefore, I am pleased to recommend its publication after addressing some minor issues.

1. Regarding the optical transparency, it is important to show the transmission spectrum in the optical range. Thus I recommend the authors add the simulation or experimental data.

2. Besides the reflection and transmission spectra, the absorption spectrum should be shown.

3. In reflection and transmission spectra, there are different resonances? What are these resonances? Please add some discussions (for instance, plotting the electromagnetic modes)

4. The authors could consider citing the following papers with similar topics to make it more comprehensive.

(1) Scientific reports 6, 39445, 2016

(2) Nanophotonics 7 (6), 1129-1156, 2018

(3) Optics Express 26 (13), 16466-16476, 2018

(4) Optics Express, 27, 28313-28323 (2019)

Author Response

To Reviewer 2

Comment:

In the paper, the authors experimentally demonstrate an optically transparent absorber with broadband absorption in the microwave region. The experimental measurements agree well with the theoretical predictions, showing broadband absorption. Though the topic has been widely investigated, the paper is still interesting. Therefore, I am pleased to recommend its publication after addressing some minor issues.

  1. Regarding the optical transparency, it is important to show the transmission spectrum in the optical range. Thus I recommend the authors add the simulation or experimental data.

Our Response:

Thank you very much for your good comments. The optical transparency of the ITO film is a intrinsic property, which has been fully and widely researched and demonstrated in many works [R6-R10]. Thus, the optical transparency is not for demonstration in this work since our main points are on the flexible absorption performance realized by the metasurface. To illustrate its characteristic, we have showed the transparency value offered by the fabricator. The transparency value of the ITO film we applied is about 88%, as we have mentioned in the last sentence in page 7.

[R6] C. Cali, M. Mosca, and G. Targia, "Deposition of indium tin oxide films by laser ablation: Processing and characterization," Solid-State Electronics, vol. 42, no. 5, pp. 877-879, May 1998, doi: 10.1016/s0038-1101(98)00084-7.

[R7] T. Maruyama and K. Fukui, "INDIUM TIN OXIDE THIN-FILMS PREPARED BY CHEMICAL VAPOR-DEPOSITION," Thin Solid Films, vol. 203, no. 2, pp. 297-302, Aug 30 1991, doi: 10.1016/0040-6090(91)90137-m.

[R8] S. H. Mohamed, F. M. El-Hossary, G. A. Gamal, and M. M. Kahlid, "Properties of Indium Tin Oxide Thin Films Deposited on Polymer Substrates," Acta Physica Polonica A, vol. 115, no. 3, pp. 704-708, Mar 2009, doi: 10.12693/APhysPolA.115.704.                

[R9] T. Nakao et al., "Characterization of indium tin oxide film and practical ITO film by electron microscopy," Thin Solid Films, vol. 370, no. 1-2, pp. 155-162, Jul 17 2000, doi: 10.1016/s0040-6090(00)00951-2.

[R10] B. Parida, H. Y. Ji, G. H. Lim, S. Park, and K. Kim, "Enhanced photocurrent of Si solar cell with the inclusion of a transparent indium tin oxide thin film," Journal of Renewable and Sustainable Energy, vol. 6, no. 5, Sep 2014, Art no. 053120, doi: 10.1063/1.4897656.

Comment:

  1. Besides the reflection and transmission spectra, the absorption spectrum should be shown.

Our Response:

Thank you very much for your good suggestion. We have supplemented the absorption spectrum and the related discussion in the revised paper. The absorption from 2GHz to 18GHz is almost from 0.7, in which the performance in high frequency is much better that it in low frequency. The detailed illustration we supplemented is provided as below

The supplemented sentences:

To clearly present the absorption performance, we calculated the normalized absorbing coefficient in Figure 2(e), when the excitation is set at port 1 (from the front side) and port 2 (from the backside) respectively. We can clearly observe that the absorption from 2GHz to 18GHz is almost from 0.7, in which the performance in high frequency is much better than it in low frequency.

Comment:

  1. In reflection and transmission spectra, there are different resonances? What are these resonances? Please add some discussions (for instance, plotting the electromagnetic modes)

Our Response:

Thank you very much for your good comments. The reflection and transmission spectra showed in the Figs. 2 and 3 are corresponding to the same element structure. Actually, the absorption realized in reflection and transmission spectra are mainly resulted from the lossy property of ITO films. When the spatial propagating EM wave interacts with the ITO metasurface structure, both reflected and transmitted wave occur the distinct energy loss. To clearly presented the resonance characteristics of the metasurface element, we provide the electric field distribution for the designed element in Fig. 2, when the excitation port is set in the front or the back. The electric field of two layers metasurface is analyzed from the front and back view.

It should be noted that we only provide the situation for the linear x-polarization incidence because the dual-layer structure of the metasurface is rotational symmetric for both side excitations. The supplemented illustration is provided as below,

The supplemented sentences:

We also show the simulated electric field distribution on the element structure when the excitation is port 1 or port 2. Both the front and the back view is exhibited in figure 2(f) and 2(g), to clearly show its resonance. When the excitation is set at port 1, the electric field on the front layer is obviously stronger than the back layer; on the contrary, when the port 2 is excited, the electric field on the backside is much stronger.

Comment:

  1. The authors could consider citing the following papers with similar topics to make it more comprehensive.

(1) Scientific reports 6, 39445, 2016

(2) Nanophotonics 7 (6), 1129-1156, 2018

(3) Optics Express 26 (13), 16466-16476, 2018

(4) Optics Express, 27, 28313-28323 (2019)

Our Response:

Thank you very much for your good suggestions. We have added the above papers into our introduction as following,

The supplemented sentences:

To address the processing limits of three-dimension (3D) structure, metasurfaces and metafilms with ultrathin profile and similar manipulation capability on EM wave are presented [R11-R13].

The operating frequency range of metamaterial absorber has covered from microwave [23], to terahertz [24] and optical domain [25, R14].

[R11] F. Ding, Y. Yang, R. A. Deshpande, and S. I. Bozhevolnyi, "A review of gap-surface plasmon metasurfaces: fundamentals and applications," Nanophotonics 2018, 7, 1129-1156.

[R12] Z. Li, D. Rosenmann, D. A. Czaplewski, X. Yang, and J. Gao, "Strong circular dichroism in chiral plasmonic metasurfaces optimized by micro-genetic algorithm," Optics Express 2019, 27, 28313-28323.

[R13] S. Zhong, L. Wu, T. Liu, J. Huang, W. Jiang, and Y. Ma, "Transparent transmission-selective radar-infrared bi-stealth structure," Optics Express 2018, 26, 16466-16476.

[R14] F. Ding, J. Dai, Y. Chen, J. Zhu, Y. Jin, and S. I. Bozhevolnyi, "Broadband near-infrared metamaterial absorbers utilizing highly lossy metals," Scientific Reports 2016, 6, 39445.